# Perception of an object's global shape is best described by a model of skeletal structure in human infants

Vladislav Ayzenberg[1]*, Stella Lourenco[2]

[1]Neuroscience Institute, Carnegie Mellon University, Pittsburgh, United States; [2]Department of Psychology, Emory University, Atlanta, United States

**Abstract** Categorization of everyday objects requires that humans form representations of shape that are tolerant to variations among exemplars. Yet, how such invariant shape representations develop remains poorly understood. By comparing human infants (6–12 months; N=82) to computational models of vision using comparable procedures, we shed light on the origins and mechanisms underlying object perception. Following habituation to a never-before-seen object, infants classified other novel objects across variations in their component parts. Comparisons to several computational models of vision, including models of high-level and low-level vision, revealed that infants' performance was best described by a model of shape based on the skeletal structure. Interestingly, infants outperformed a range of artificial neural network models, selected for their massive object experience and biological plausibility, under the same conditions. Altogether, these findings suggest that robust representations of shape can be formed with little language or object experience by relying on the perceptually invariant skeletal structure.

## Editor's evaluation

This well-conducted study uses relatively large sample sizes, comprehensive statistical testing, and state-of-the-art modeling to provide novel evidence that human infants generalize shape from single examples on the basis of the "shape skeleton", a structural description of the part structure of the shape. It will be of interest to researchers working on object shape processing and on the development of visual perception.

*For correspondence: vayzenb@cmu.edu

Competing interest: The authors declare that no competing interests exist.

## Introduction

The appearance of objects within a single category can vary greatly. For instance, despite the shared category of dog, the exemplars (i.e. different dogs) may have different snouts, tails, and/or torsos. Despite this variability, humans readily categorize never-before-seen dog breeds as members of the same basic-level category. How do we do this? Although there is widespread agreement that shape information is crucial for object categorization (*Biederman, 1995*; *Mervis and Rosch, 1981*), it remains unclear how humans come to form global representations of shape that are tolerant to variations among exemplars.

It has been suggested that global shape information becomes crucial for object categorization in early childhood because linguistic experience, particularly the learning of object labels (e.g. 'dog'), draws children's attention to global shape as a diagnostic cue—inducing a so-called 'shape bias' (*Landau et al., 1998*; *Smith et al., 1996*; *Smith et al., 2002*). Indeed, labels may bootstrap object recognition abilities more generally, such that even prelinguistic children are better at individuating and categorizing objects when verbal labels are provided (*Ferry et al., 2010*; *Xu et al., 2005*). The

advantage of labeled object experience is particularly evident in supervised artificial neural networks (ANNs), which have begun to match the object recognition abilities and neural representations of human adults (*Krizhevsky et al., 2017*; *Rajalingham et al., 2018*; *Schrimpf et al., 2018*). These models learn the diagnostic properties of objects following training with millions of labeled natural-istic images. With appropriate experience, these models may even develop a shape bias that supports object categorization, at least when generalizing across variations in color or texture (*Ritter et al., 2017*; *Tartaglini et al., 2022*). Thus, global representations of shape may develop with labeled object experience that highlights the diagnostic properties of objects.

An alternative possibility, however, is that rather than labeled experience, humans develop global shape representations by relying on (non-linguistic) invariant perceptual properties inherent to the object (*Biederman, 1987*; *Feldman, 1997*; *Rakison and Butterworth, 1998*; *Sloutsky, 2003*). One such property is known as the shape skeleton—a quantitative model that describes an object's global shape via a series of internal symmetry axes (*Blum, 1967*; *Feldman and Singh, 2006*). These axes define the topological arrangement of object parts, making models of skeletal structure tolerant to local variations in shape typical of basic-level exemplars (*Ayzenberg et al., 2019a*; *Wilder et al., 2011*). From this perspective, extensive experience with objects and linguistic labels may not be necessary to form global shape representations, and, instead, one might rely on the shape skeleton (*Feldman, 1997*). However, the contributions of labeled experience and the shape skeleton are diffi-cult to examine because, by adulthood, humans have had massive amounts of labeled object experi-ence, making the source of their shape representations ambiguous. Here we tested whether human infants (who have little linguistic or object experience) represent global object shape according to a shape skeleton.

Object representations in infancy have most often been tested using visual attention procedures. In these experiments, infants are typically habituated to stimuli that share a common visual dimension (e.g. shape), but vary according to other dimensions (e.g. color). Infants are then tested with objects that are either familiar (e.g. similar in shape to the habituated object) or novel (i.e. different in shape to the habituated object). If infants learn the relevant dimension during the habituation phase, then their looking times are longer for the novel object compared to the familiar one. Habituation para-digms provide an informative window into infants' object representations because they reveal what properties infants learned during the habituation phase and subsequently generalized to the test phase. Using this approach, researchers have shown that newborns can already discriminate between simple 2D shapes (*Slater et al., 1983*) and display shape constancy, such that they recognize a shape from a novel orientation (*Slater and Morison, 1985*). By 6 months of age, infants' shape representa-tions are also robust to variations among category exemplars, such that they can categorize objects using only the stimulus' shape silhouette (*Quinn et al., 1993*; *Quinn et al., 2001a*), as well as extend category membership to objects with varying local contours, but the same global shape (*Quinn et al., 2002*; *Quinn et al., 2001b*; *Turati et al., 2003*). However, the mechanisms underlying global shape representation remain unclear. Indeed, because infants in these studies were habituated to multiple (often familiar) objects, it is unclear whether shape representations in these studies were learned from the statistics of the habituation period (*Oakes and Spalding, 1997*; *Younger, 1990*) or, rather, are an invariant perceptual property infants extract from objects more generally.

In the current study, we used a habituation paradigm to examine how 6–12 months old infants represent the global shape of objects. In particular, we tested whether infants classified never-before-seen objects by comparing the similarity between their shape skeletons. Importantly, we habitu-ated infants to only a single object so as to measure their pre-existing shape representations, rather than ones they may have learned over the course of habituation. We chose to test 6–12-month-olds because the visual experience at this age is dominated by just a few common objects (~10 objects; *Clerkin et al., 2017*) and they have relatively little linguistic understanding (~7 words; *Bergelson and Swingley, 2012*). Moreover, we elucidate the mechanisms that support object perception in infancy by comparing infants to a range of computational models. Of particular interest was a flux-based medial axis algorithm that computes a shape skeleton from images (*Rezanejad and Siddiqi, 2013*). If infants represent objects using the shape skeleton, then their classification judgments should best match that of a Skeletal model.

In addition, we included four ANNs, known for their success on object recognition tasks but which do not represent the shape skeleton. These ANNs included two ResNet models, one trained on

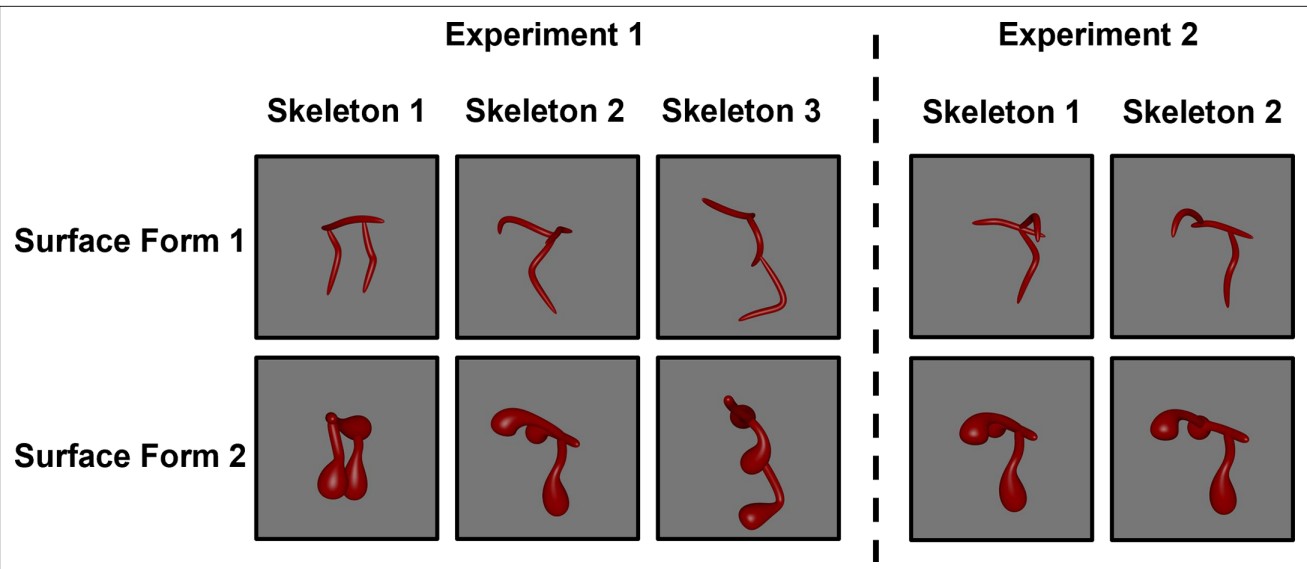

**Figure 1.** Screen shots of the stimuli used in Experiment 1 (left) and Experiment 2 (right). Objects were presented as rotating videos during habituation and test phases.

ImageNet (ResNet-IN), a convolutional ANN frequently used in object recognition tasks, and a variation of ResNet trained on Stylized-ImageNet (ResNet-SIN), an image set that leads models to develop a shape bias, at least in relation to color and textures cues (*Geirhos et al., 2018*). Two other ANNs were CorNet-S, a top-performing model of object recognition behavior and neural processing in primates, as measured by the brain-score benchmark (*Schrimpf et al., 2018*), and ResNext-SAY, a model trained with an unsupervised learning algorithm on first-person videos from infants (*Orhan et al., 2020*). All of these ANNs were included because they exhibit varying degrees of biological plausibility in terms of neural organization or visual experience (*Russakovsky et al., 2015*; *Schrimpf et al., 2018*). If infant performance is best matched by ANNs, then this would suggest that global shape representations might be learned as a diagnostic cue following extensive object experience. Importantly, because none of these models represent the shape skeleton, they make for an excellent contrast to the Skeletal model. Finally, we also included a model of pixel similarity, and FlowNet, a model of optic flow (*Ilg et al., 2017*) in order to assess the extent to which shape representations may be supported by lower-level visual properties like image similarity (*Kiat et al., 2022*; *Xie et al., 2021*) or motion trajectory (*Kellman, 1984*; *Kellman and Short, 1987*; *Wood and Wood, 2018*). Altogether, these comparisons provided a novel approach to understanding object perception in human infants.

Because the strength of any object classification task depends on the degree of dissimilarity between training (i.e. habituation) and test objects, infants were tested with objects that mimicked the variability of basic-level category exemplars (*Figure 1*). Within-category objects comprised objects with the same skeletons, but visually distinct component parts. Variation in the component parts was generated by manipulating the objects' surface forms, thereby changing both the image-level features and the non-accidental properties (NAPs; *Biederman, 1987*), without altering the skeleton. Between-category objects comprised different skeletons and surface forms. If infants are capable of classifying objects vis-à-vis a shape skeleton, then they should look longer at objects with different skeletons compared to those with the same skeleton, even though both objects differ from the habituated object in surface form. Importantly, we tested whether within-category and between-category objects were equally discriminable by infants to ensure that any differences in looking time were not related to infants' ability to differentiate surface forms. Moreover, if infants rely on objects' shape skeletons to determine similarity, then their classification performance should be best matched by the Skeletal model, rather than ANNs, or other models of vision. Thus, the current study provides critical insight regarding the nature of shape representations at an age when experience with labeled objects is minimal, and, crucially, provides a novel benchmark by which to evaluate the biological plausibility of computational models of vision.

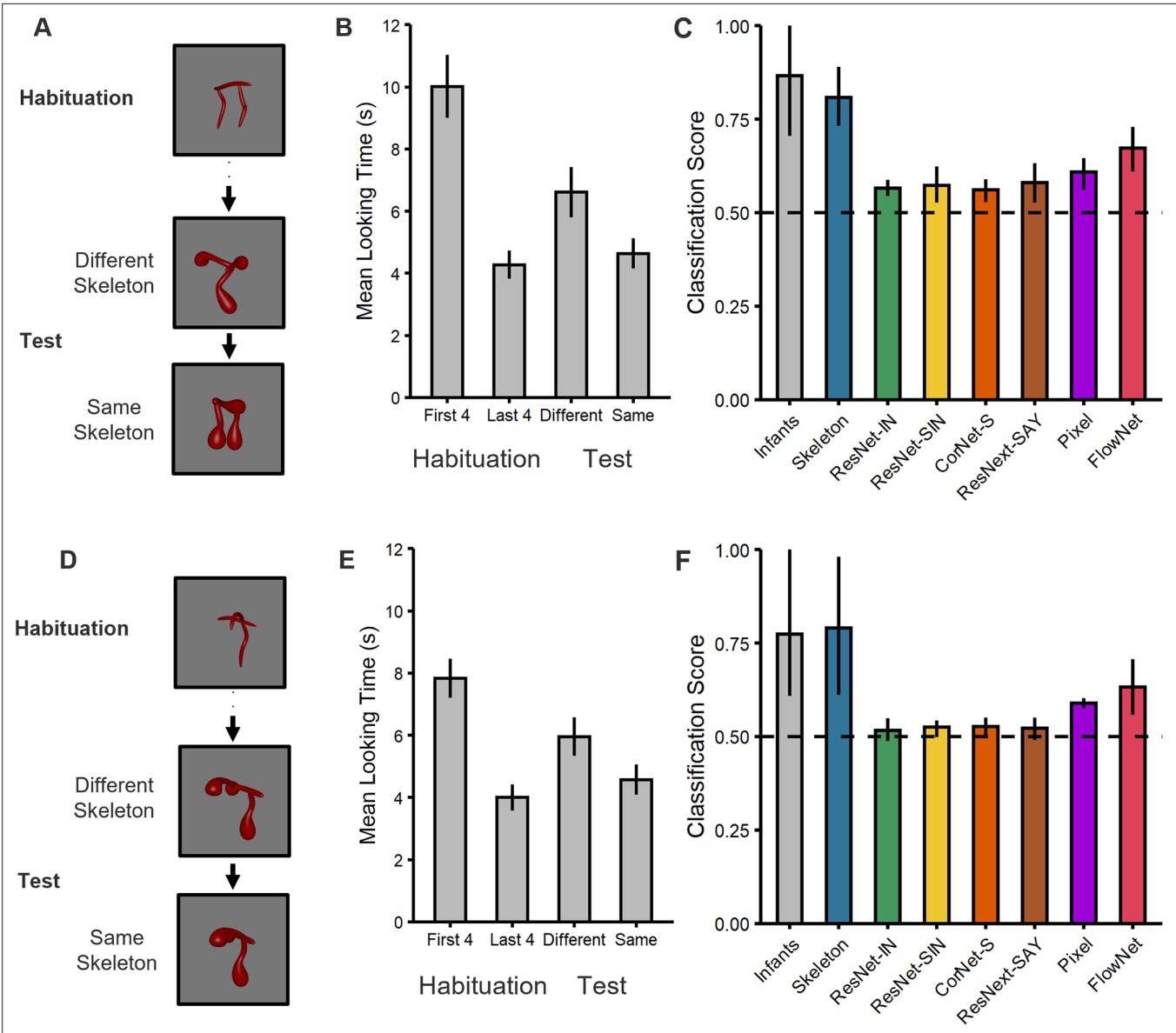

**Figure 2.** Experimental design and results for (top) Experiment 1 and (bottom) Experiment 2. (**A, D**) Illustration of the experimental procedure administered to infants and the computational models in (**A**) Experiment 1 and (**D**) Experiment 2. Infants and models were habituated to one object and then tested with objects that consisted of either the same or different shape skeleton. Both types of test objects (counterbalanced order) differed in their surface forms from the habituation object. (**B, E**) Mean looking times for (**B**) Experiment 1 and (**E**) Experiment 2. For the habituation phase, results are shown for the first four and last four trials. For the test phase, results are shown for the two types of test objects (i.e. same and different skeletons; 3 test trials each). Error bars represent SE. (**C, F**) Classification performance for infants and models for (**C**) Experiment 1 and (**F**) Experiment 2. Error bars represent bootstrapped confidence intervals, and the dashed line represents chance performance.

## Results

Infants' looking times were analyzed using two-sided paired sample $t$-tests ($\alpha$=0.05) and standard measures of effect size (Cohen's $d$). Furthermore, to ensure that sample size decisions did not unduly influence the results, we also conducted non-parametric (Binomial tests) and Bayesian analyses. A Bayes factor ($BF_{10}$) was computed for each analysis using a standard Cauchy prior ($d$=0.707). A $BF_{10}$ greater than 1 is evidence that two distributions are different from one another, whereas a $BF_{10}$ less than 1 is evidence that two distributions are similar to one another (***Rouder et al., 2009***).

## How do infants represent shape?

A comparison of the two types of test trials in Experiment 1 (*Figure 2A*) revealed that, across the test phase, infants looked longer at the object with the different skeleton compared to the one with the matching skeleton ($t$[33]=3.04, p=0.005, $d$=0.52, 95% CI [0.16, 0.88], $BF_{10}$=8.42; *Figure 2B*), with the majority of infants showing this effect (25/34 infants, p=0.009). In addition, a comparison between the end of habituation (mean of last 4 trials) and looking times across the test phase revealed that dishabituation only occurred for the object with the different skeleton ($t$[33]=3.36, p=0.002, $d$=0.58, 95% CI [0.21, 0.94], $BF_{10}$=17.47; 25/34 infants, p=0.009), not the object with the matching skeleton ($t$[33]=1.00, p=0.325, $d$=0.17, 95% CI [–0.16, 0.51], $BF_{10}$=0.29; *Figure 2B*). Thus, infants treated objects with matching skeletons as more similar to one another than objects with different skeletons.

However, one might ask whether infants were simply unable to differentiate between surface forms, leading to greater looking at the object with a different skeleton. To test this possibility, we compared infants' looking times on the first test trial following habituation to the last trial during habituation. Because the first test trial immediately follows habituation, this comparison allows for a direct measure of perceptual discriminability between habituation and test objects when the memory demands are minimal and in the absence of carry-over effects between test objects. This analysis revealed that infants were indeed capable of discriminating objects on the basis of surface form alone. That is, infants dishabituated to the object with the same skeleton but different surface form, $t$(15) = 3.76, $p$=0.002, $d$=0.94, 95% CI [0.34, 1.52], $BF_{10}$=22.23, with the majority of infants showing this effect (14/16, $p$=0.004). Moreover, and crucially, infants' looking times on the first test trial did not differ for either test object (8.31 s vs. 9.18 s; $t$[32]=0.50, $p$=0.624, $d$=.17, 95% CI [–0.51, 0.84], $BF_{10}$=0.36). These findings demonstrate that not only could infants differentiate between surface forms, but also, that the two types of test objects were matched for discriminability relative to the habituation object. These findings argue against a pure discrimination account and, instead, support the interpretation that infants classified objects on the basis of skeletal similarity.

But did infants actually rely on the object's shape skeleton to judge similarity or some other visual representation? To explore this possibility, we compared infants' classification behavior to a flux-based Skeletal model, a range of ANNs (ResNet-IN, ResNet-SIN, and CorNet-S, ResNext-Say), and models of image similarity (Pixel) and motion (FlowNet). If infants relied on the shape skeleton to classify objects, then their performance would be best matched by the Skeletal model, rather than the others.

Models were tested with the same stimuli presented to infants using a procedure comparable to the habituation paradigm. More specifically, because habituation/dishabituation can be conceived as a measure of alignment between the stimulus and the infant's internal representation (*Mareschal et al., 2000*; *Westermann and Mareschal, 2004*), we tested models by feeding their outputs into an autoencoder and measuring the error signal across habituation and test phases (see Methods). Like habituation paradigms, the error signal of an autoencoder reflects the degree of alignment between the internal representation of the model and the input stimulus (for review, see *Yermolayeva and Rakison, 2014*). Unlike conventional classifiers, which often require multiple labeled contrasting examples (e.g. Support Vector Machines), autoencoders allow for measuring a model's performance following exposure to just one exemplar, as with infants in the current study. Moreover, like infant learning during habituation, the learned representation of an autoencoder reflects the entire habituation video, rather than the representation of individual frames. Most importantly, unlike other techniques, autoencoders can be tested using the same habituation and test criteria as infants (see Methods). For comparison, performance for both infants and models was converted into a classification score (see Methods) and significance was assessed using bootstrapped confidence intervals (5000 iterations).

These analyses revealed that all models performed above chance (0.50; see *Figure 2C*). However, and importantly, infant performance was best matched to the Skeletal model. Both infants and the Skeletal model performed significantly better than all other models, except FlowNet. That infants outperformed the ANNs suggests that extensive object experience may not be necessary to develop robust shape representations. Likewise, that infants outperformed the Pixel model suggests that infant performance is not explained by the low-level visual similarity between objects. However, the success of FlowNet does suggest that motion information may contribute to representations of shape. Nevertheless, FlowNet's performance did not differ from that of any ANN or the Pixel model, leaving its exact role in object classification unclear. Altogether, these results suggest that infants' performance is

most closely aligned with the Skeletal model and, thus, that infants classified objects, at least in part, by relying on the similarity between objects' shape skeletons.

## Can infant performance be explained by another representation of global shape?

An alternative explanation for the findings from the first experiment is that, rather than the shape skeleton, infants classified objects using another representation of global shape—namely, the coarse spatial relations among object parts (*Biederman and Gerhardstein, 1993*; *Hummel and Stankiewicz, 1996*). Whereas a Skeletal model provides a continuous, quantitative description of part relations, a model based on coarse spatial relations describes part relations in qualitative terms (e.g. two parts below a third vs. two parts on either side of a third). In Experiment 1, test objects with different skeletons also consisted of part relations that could be considered qualitatively different from that of the habituated object, making it unclear whether infants relied on coarse spatial relations instead of the shape skeletons.

To address this possibility, in Experiment 2, coarse spatial relations were held constant between habituation and test phases (i.e. two parts on either side of a third; *Figures 1 and 2D*). Thus, if infants relied on coarse spatial relations for object classification, then they would fail to dishabituate to the test object that differed in the shape skeleton (but not coarse spatial relations). However, we found that infants continued to look longer at the object with the different skeleton compared to the one with the matching skeleton ($t$[47]=2.60, $p$=0.012, $d$=0.38, 95% CI [0.08, 0.67], BF$_{10}$=3.18; *Figure 2E*), with the majority of infants showing this effect (33/48 infants, $p$=0.013). Moreover, infants only dishabituated during the test phase to the object with the different skeleton ($t$[47]=3.63, p<0.001, $d$=0.52, 95% CI [0.22, 0.82], BF$_{10}$=39.77; 36/48 infants, $p$<0.001), not the one with the matching skeleton ($t$[47]=1.42, $p$=0.163, $d$=0.16, 95% CI [–0.08, 0.49], BF$_{10}$=0.40), as in Experiment 1. The results from this experiment rule out the possibility that infants classified objects on the basis of their coarse spatial relations, rather than their shape skeletons.

Importantly, as in the previous experiment, we also compared infants' looking times on the first test trial to the last trial during habituation to ensure that infants could distinguish surface forms and that the two test objects were equally discriminable. We found that infants dishabituated to the object with the same shape skeleton but different surface form, $t$(22)=3.51, $p$=0.002, $d$=0.73, 95% CI [0.26, 1.19], BF$_{10}$=19.71, with the majority of infants showing this effect (17/23, p=0.035), suggesting that discrimination was possible on the basis of surface form alone. Moreover, infants' looking times on the first test trial did not differ for the two types of test objects (6.87 s vs. 10.83 s; $t$[45]=1.47, p=0.149, $d$=0.43, 95% CI [–0.15, 1.00], BF$_{10}$=0.69), confirming comparable discriminability. Thus, as in Experiment 1, these findings suggest that although infants were capable of discriminating both types of test objects, they nevertheless treated objects with the same skeletons as more similar to one another.

To ensure that the effects in Experiment 2 were not unduly influenced by the larger sample size, we computed bootstrapped CIs on a smaller sample. For each bootstrap procedure (10,000 iterations), we calculated Cohen's $d$ on data that were resampled (without replacement) to match the sample size of Experiment 1 ($n$=34). We found that infants dishabituated to the test object with a different skeleton, 95% CI [0.22, 0.87], but not the same skeleton, 95%CI [–0.13, 0.61]. Infants also looked longer at the test object with a different skeleton than the test object with the same skeleton, 95% CI [0.20, 0.61], confirming the robustness of these effects regardless of sample size. Finally, analyses of the first test trial confirmed that infants discriminated between surface forms when objects had the same skeletons, 95% CI [0.22, 0.81,] and that looking times did not significantly differ between the two types of test trials, 95% CI [–0.15, 0.49].

To examine whether infants' performance was best described by a Skeletal model we, again, compared infants to a model that represents the shape skeleton, as well as to models that do not. These analyses revealed that the Skeletal, Pixel, and FlowNet models all performed above chance (*Figure 2F*), but infant performance was most closely matched to the Skeletal model. None of the ANNs performed above chance. As in Experiment 1, both infants and the Skeletal model significantly outperformed the ANNs and the Pixel model, but not FlowNet. These findings again suggest that infants' judgments reflect the use of skeletal structure and that there may be a role for motion information in this ability.

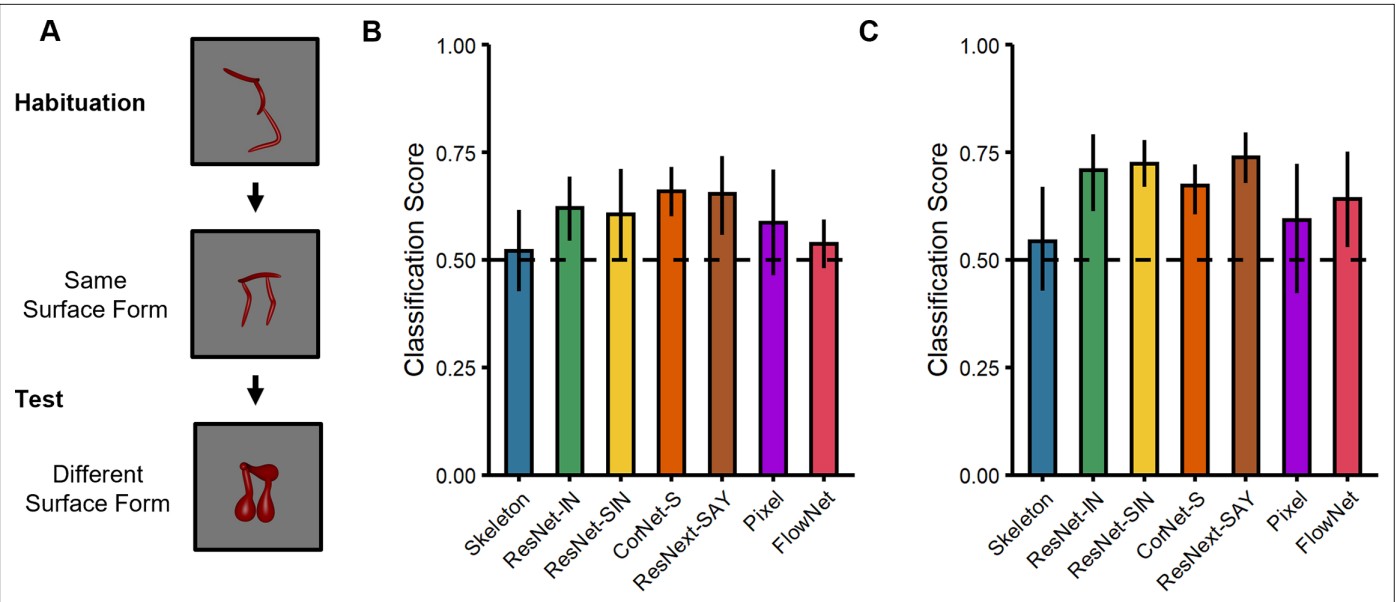

**Figure 3.** Experimental design and results for the surface form classification task used with the computational models. (**A**) Illustration of the experimental procedure administered to models. (**B–C**) Classification performance of models on stimuli from (**B**) Experiment 1 and (**C**) Experiment 2. Error bars represent bootstrapped confidence intervals and dashed lines represent chance performance.

## Model classification using local shape properties

Given the competitive performance of ANNs on other object recognition tasks, and their match to adult performance in other studies (*Schrimpf et al., 2018*), one might wonder why ANNs performed so poorly on our task. One possibility is that ANNs rely on local shape properties (e.g. surface form), which were irrelevant to our task (*Baker et al., 2018*; *Baker et al., 2020*). To test this possibility, we examined whether ANNs were capable of classifying objects using local shape properties, namely, the surface forms of the objects. For comparison, we also tested the other models' performance on this task (Skeleton, Pixel, and FlowNet).

Using a comparable procedure to the previous experiments, all models were tested with objects in which the surface forms either matched or differed from the habituated object; both test objects differed in their shape skeletons (*Figure 3A*). For objects from Experiment 1 (*Figure 1*), the analyses revealed that ResNet-IN, CorNet-S, and ResNext-SAY performed above chance (Ms = 0.62–0.66, 95% CIs [0.53–0.59, 0.70–0.76]; *Figure 3B*). By contrast, none of the other models (Skeleton, ResNet-SIN, Pixel, FlowNet) performed differently from chance (Ms = 0.52–0.61, 95% CIs [0.40–0.48, 0.60–0.72]; *Figure 3B*) when classifying objects by surface form (i.e. across shape skeletons). For objects from Experiment 2 (*Figure 1*), all models performed above chance (Ms = 0.67–0.74, 95% CIs [0.51–0.67, 0.72–0.80]), except Skeletal and Pixel models (Ms = 0.54–0.59, 95% CIs [0.40–0.41, 0.68–0.73]; *Figure 3C*). Altogether, these findings demonstrate that most ANNs are capable of classifying objects using local shape properties. These findings are consistent with other research showing sensitivity to local shape properties in ANNs (*Baker et al., 2018*).

## General discussion

Although it is well known that shape is crucial for object categorization, much speculation remains about how representations of global shape are formed in development. Here, we demonstrate that infants represent global shape by extracting a skeletal structure. With only one exemplar as a reference, infants classified objects by their shape skeletons across variations in local properties that altered component parts of the objects, including when coarse spatial relations could not be used as a diagnostic cue. Moreover, a Skeletal model provided the closest match to infants' performance, further suggesting that infants relied on the shape skeleton to determine object similarity. Based on these findings, we would argue that the formation of robust shape representations in humans is largely rooted in sensitivity to the shape skeleton, an invariant representation of global shape.

## The role of other visual properties in object perception

It is important to acknowledge that our results also suggest that object classification can be accomplished on the basis of image-level similarity. In particular, when generalizing across the surface form, a Pixel model performed above chance. Does this mean that infants' performance can be explained by low-level image similarity between habituation and test objects rather than the shape skeleton? We would suggest not. First, we found that infants discriminated both test objects from the habituation object equally, which argues against the possibility that infants' performance was strictly based on differences in image-level similarity. Second, both infants and the Skeletal model outperformed the Pixel model in both experiments, further arguing against an account based on image-level similarity alone. Nevertheless, we would not argue that image similarity plays no role in object perception, particularly in infants. Indeed, recent studies comparing the visual representations of infants and computational models reveal that low-level visual similarity explains more variance in infants' behavioral and neural responses than the upper layers of ANNs (*Kiat et al., 2022*; *Xie et al., 2021*). Moreover, recent studies suggest that object categorization in infancy may be supported by the representations of the early visual cortex (V1-V3), rather than the higher-level ventral cortex, as in adults (*Spriet et al., 2022*).

Like image-level representations, shape skeletons, themselves might be an emergent property of early visual regions. Specifically, several studies have suggested that skeletal representations may emerge in area V3 as a consequence of between-layer recurrent interactions among border-ownership cells in area V2 and grouping cells in area V3 (*Ardila et al., 2012*; *Craft et al., 2007*). Interestingly, representations of skeletal structures in V3 are invariant across changes in component parts, which lends support to their role in categorization (*Ayzenberg et al., 2022*; *Lescroart and Biederman, 2013*). Other research on the anatomical and functional organization of V3 in primates shows evidence of functional maturity shortly after birth (*Arcaro and Livingstone, 2017*; *Ellis et al., 2020*; *Wiesel and Hubel, 1974*), further raising the possibility that skeletal structure is represented in this area with little object experience or any language.

In the present study, we also found that FlowNet, a model of optic flow, appeared comparable to infants and the Skeletal model at classifying objects when they differed in surface form. This finding highlights the role of motion in the formation of invariant object representations (*Lee et al., 2021*; *Ostrovsky et al., 2009*). It is known that infants are better at recognizing objects from novel viewpoints when they are first familiarized with moving objects rather than static images (*Kellman, 1984*; *Kellman and Shipley, 1991*; *Kellman and Short, 1987*). Moreover, controlled-rearing studies with chicks have found that viewpoint-invariant object recognition occurs only if the chicks are raised in environments in which objects exhibit smooth, continuous motion (*Wood and Wood, 2018*). Indeed, several studies suggest that motion information may initially bootstrap infants' ability to extract 3D shape structure (*Kellman and Arterberry, 2006*; *Ostrovsky et al., 2009*; *Wood and Wood, 2018*). Thus, motion information may work in concert with shape skeletons to support object perception in infancy. Yet, it is important to note that classifying objects on our task was also possible without relying on motion cues (*Quinn et al., 2001a*; *Quinn et al., 2001b*). Indeed, the Skeletal model, which does not incorporate any motion information, was more closely aligned to the performance of human infants than was FlowNet.

In contrast to infants and the other models, ANNs were not capable of classifying objects across variations in surface form. However, they were capable of classifying objects by their surface form (across variation in shape skeleton), ruling out alternative explanations for their poor performance in the first task based on idiosyncrasies of the stimuli or testing procedures. Moreover, this finding suggests that, regardless of the specific architecture or training type, conventional ANNs rely on qualitatively different mechanisms than do humans to represent objects. This class of models may be especially sensitive to local object properties, making them susceptible to spurious changes in the objects' contours (*Baker et al., 2018*; *Baker et al., 2020*). By contrast, representations of skeletal structure in humans are particularly robust to perturbations (*Ayzenberg et al., 2019a*; *Feldman and Singh, 2006*) and, thus, may be especially well-suited to basic-level categorization. Although conventional ANNs (e.g. ResNet-SIN) can generalize across variations in color and texture (*Geirhos et al., 2018*; *Tartaglini et al., 2022*), fundamental changes to ANNs' architectures and/or training may be needed before they can achieve human-like categorization on the basis of global shape.

Despite our claim that infants relied on skeletal similarity to classify objects, we would not suggest that it is the only information represented by humans. Indeed, by adulthood, humans also use visual properties such as texture (*Jagadeesh and Gardner, 2022*; *Long et al., 2018*) and local contours (*Davitt et al., 2014*), as well as inferential processes such as abstract rules (*Ons and Wagemans, 2012*; *Rouder and Ratcliff, 2016*) and the object's generative history (*Fleming and Schmidt, 2019*; *Spröte et al., 2016*) to reason about objects. Properties such as texture and local features may even override shape skeletons in certain contexts, such as when identifying objects in the periphery (*Gant et al., 2021*) or during subordinate-level categorization (*Davitt et al., 2014*; *Tarr and Bülthoff, 1998*). We suggest that the shape skeleton may be uniquely suited to the basic level of categorization, wherein objects have similar global shapes but vary in their component parts (*Biederman, 1995*; *Rosch et al., 1976*). However, more research will be needed to understand the extent to which children may also make use of other properties when categorizing objects.

## Implications for one-shot categorization

A key aspect of our design was that infants were habituated to a single, novel object. We did this to better understand the pre-existing visual representations infants rely on for object perception, rather than the ones they may learn over the course of habituation. Using this approach, we found that infants reliably classified objects on the basis of the shape skeleton. Might this result also suggest that infants are capable of one-shot categorization using the shape skeleton? One-shot categorization is the process of learning novel categories following exposure to just one exemplar (*Lake and Piantadosi, 2019Lake et al., 2011*; *Morgenstern et al., 2019*; *Shepard, 1987*). We cannot answer this question definitively, given that visual attention paradigms make it difficult to distinguish between category learning per se and judgments of visual similarity. However, it is intriguing to consider whether the shape skeleton may support rapid object learning at an age when linguistic and object experience is minimal.

How well do our results align with the one-shot categorization literature from older children and adults? On the one hand, our results are consistent with studies showing that one-shot categorization of objects by older children and adults involves identifying invariant visual properties of the objects, namely shape (*Biederman and Bar, 1999*; *Feldman, 1997*; *Landau et al., 1988*). Moreover, research using simple visual objects (e.g. handwritten characters) suggests that a skeleton-like compositional structure is central to one-shot categorization (*Lake et al., 2011*; *Lake et al., 2015*). On the other hand, our results would suggest that such categorization is possible at a much earlier age than has previously been suggested. Indeed, the extant research with both children (*Landau et al., 1998*; *Smith et al., 2002*; *Xu and Kushnir, 2013*) and adults (*Coutanche and Thompson-Schill, 2014*; *Rule and Riesenhuber, 2020*) has been taken as evidence that such categorization abilities are accomplished by leveraging extensive category and linguistic knowledge. Thus, our results are consistent with the hypothesis that one-shot category learning may be possible early in development by relying on a perceptually invariant skeletal structure.

However, we would not suggest that object experience is irrelevant in infancy. As mentioned previously, experience with moving objects may play a role in supporting the extraction of a 3D shape structure. Moreover, although infants' visual experience is dominated by a small number of objects, this experience includes a large volume of viewpoints for each object (*Clerkin et al., 2017*). Interestingly, infants' chosen views of objects are biased toward planar orientations (*James et al., 2014*; *Slone et al., 2019*), which may serve to highlight the shape skeleton, though it is unclear whether such a bias is a consequence or a cause of skeletal extraction. Moreover, certain visual experiences, such as low visual acuity at birth, may be particularly important for highlighting global shape information (*Cassia et al., 2002*; *Ostrovsky et al., 2009*). Indeed, ANNs trained with a blurry-to-clear visual regimen showed improved performance for categories where global information is important, such as faces (*Jang and Tong, 2021*; *Vogelsang et al., 2018*).

Altogether, our work suggests that infants form robust global representations of shape by extracting the object's shape skeleton. With limited language and object experience, infants generalized across variations in local shape properties to classify objects—a feat not matched by conventional ANNs. Nevertheless, by comparing infants' performance to existing computational models of vision, the present study provides unique insight into humans' representations of shape and their capacity for categorization, and may serve to inform future computational models of human vision.

# Materials and methods

## Participants

A total of 92 full-term infants participated in this study. Ten infants were excluded (Experiment 1: 5 for fussiness and 1 because of equipment failure; Experiment 2: 3 for fussiness and 1 because of equipment failure). The final sample included 34 infants in Experiment 1 (*M*=9.53 months, range = 6.47–12.20 months; 18 females) and 48 infants in Experiment 2 (*M*=9.12 months, range = 6.17–12.00 months; 20 females). Each infant was tested only once. All families gave informed consent according to a protocol approved by the Emory University Institutional Review Board (IRB) under the project 'Spatial Origins' (Study Number IRB0003452).

The sample size for Experiment 1 was determined using a priori power analysis with a hypothesized medium effect size (*d*=0.50; 1 - *β*>.8). For Experiment 2, we hypothesized that objects with the same coarse spatial relations would be more difficult to discriminate because the shape skeletons were more similar to one another (compared to Experiment 1), leading to an attenuated effect. Accordingly, to retain adequate power, we tested 14 more infants than in Experiment 1, the exact number of which was determined according to a fully counterbalanced design. Importantly, we nevertheless find the same results in Experiment 2 when the sample size is subsampled (n=34) to match that of Experiment 1.

## Stimuli

For Experiment 1, six videos of 3D novel objects were rendered from the stimulus set created by *Ayzenberg and Lourenco, 2019b*; *Figure 1*. The objects were comprised of three distinct skeletons selected (from a set of 30) on the basis of their skeletal similarity. The skeletal similarity was calculated in 3D, object-centered, space as the mean Euclidean distance between each point on one skeleton and the closest point on the second skeleton following maximal alignment. A *k*-means cluster analysis (*k*=3) was used to select three distinct skeletons, one from each cluster (*Figure 1*). We ensured that the three skeletons were matched for discriminability by analyzing participants' discrimination judgments using data from *Ayzenberg and Lourenco, 2019b*. Adult participants (*n*=42) were shown images of two objects (side-by-side) with either the same or different skeletons (same surface form). Participants were instructed to decide whether the two images showed the same or different objects. A repeated-measures ANOVA, with skeleton pair as the within-subject factor, revealed that the three skeletons used in Experiment 1 did not significantly differ in their discriminability, $F(2, 64)=0.11$, $p=0.898$.

For Experiment 2, we selected one object from Experiment 1 whose skeleton could be altered without changing the coarse spatial relations. We altered the object's skeleton by moving one segment 50% down the length of the central segment (*Figure 1*).

Each object was also rendered with two different surface forms, which changed the component parts and image-level properties of the object without altering its skeleton (*Figure 1*). The selection of these surface forms was based on adult participants' data from the study of *Ayzenberg and Lourenco, 2019b*. In a match-to-sample task, participants (*n*=39) were shown one object (sample) placed centrally above two choice objects. One of the choice objects matched the sample's skeleton, but not the surface form, and the other choice object matched the sample's surface form, but not the skeleton. Participants were instructed to decide which of the two choice objects was most likely to be from the same category as the sample object. Participants performed worst at categorizing objects by their skeleton when surface form 1 was paired with surface form 2, *M*=0.58, compared to the other surface forms (*Ms* = 0.61–0.78). Thus, by choosing the surface forms that presented adult participants with the greatest conflict, we provided infants with an especially strong test of generalization on the basis of the skeletal structure.

In a separate set of analyses, we tested whether the surface forms were comprised of qualitatively different component parts by having participants rate each surface form on the degree to which it exhibited a specific non-accidental property (NAP). During a training phase, adult participants (*n*=34) were taught four NAPs (drawn from *Amir et al., 2012*). They then rated the degree to which each surface form exhibited a particular NAP. The four NAPs were: (1) *taper*, defined as whether the thickness of an object part was reduced towards the end; (2) *positive curvature*, defined as whether an object part bulged outwards; (3) *negative curvature*, defined as the degree to which an object part caved inwards; and (4) *convergence to vertex*, defined as whether an object part ended in a point. Prior to the statistical analyses, we ensured that all participants in the sample exhibited

reliable performance (αs >0.7). A repeated-measures ANOVA, with NAP as the within-subject factor and surface form as the between-subject factor, revealed a significant main effect of surface form, $F$(1, 66)=64.00, $p$<0.001, such that surface forms corresponded to different NAPs. Thus, because objects between habituation and test phases consisted of different NAPs, it could not be used as a diagnostic cue for categorizing the different test objects.

We also tested whether objects with different surface forms, but the same skeleton, had significantly different image-level properties, in order to ensure that both objects presented to infants during the test phase differed from the habituation object in these properties. Each object video was converted into a sequence of images (30 frames/s; 300 frames total), which were analyzed with the Gabor-jet model (*Margalit et al., 2016*). This model overlays a 12 × 12 grid of Gabor filters (5 scales × 8 orientations) on each image. The image is convolved with each filter, and the magnitude and phase of the filtered image is stored as a feature vector. Paired *t*-tests were used to compare the feature vectors from each frame of one video to the corresponding frames of the second video. To provide an estimate of the image-level difference across the entire video, the resulting p-values from each *t*-test were then averaged across frames. This analysis revealed that objects with different surface forms (but same skeleton) had significantly different image-level properties (*p*=0.002), whereas objects with different skeletons (but same surface forms) did not (*p*=0.090). In other words, the surface forms used in the present study were actually more variable than the shape skeletons with respect to image-level properties.

Finally, we ensured that surface forms were matched in discriminability to the selected skeletons. Adult participants (*n*=41) conducted a surface form discrimination task, wherein they were shown images of two objects (side-by-side) which consisted of either the same or different surface forms (same skeleton). Participants were instructed to decide whether the two images showed the same or different objects. Participants discriminated between surface forms 1 and 2 significantly better than would be predicted by chance, *M*=0.86, *t*(40) = 8.95, *p*<.001, and importantly, discrimination accuracy between surface forms did not differ from discrimination accuracy between skeletons, *t*(80) = 0.02, *p*=.981.

## Procedure for testing infants

Each infant sat on their caregiver's lap approximately 60 cm from a 22-inch computer monitor (1920×1080 px). Caregivers were instructed to keep their eyes closed and to refrain from interacting with the infant during the study session. The experiment was controlled by a custom program written in Visual Basic (Microsoft) and gaze data were recorded with an EyeLink 1000 plus eye tracker recording at 500 Hz (SR-Research). Prior to the start of the experiment, the eye tracker was calibrated for each infant using a 5-point calibration routine. Looking times were coded as any fixation falling within the screen for at least 500 ms. Any trial failing to meet this criterion was not analyzed (2.03% of trials in Experiment 1; 2.04% of trials in Experiment 2).

The experiment consisted of a habituation phase, in which infants were presented with one object, followed by a test phase where classification was tested using objects with matching and different skeletons. Both test objects differed from the habituated object in their surface form (see *Figure 2A and D*). Each trial began with an attention-getting stimulus, which remained onscreen until infants fixated on it for 2 s. On each trial, infants were then shown a video of a single object rotating back-and-forth across 60° (12° per second). Each video remained on screen for 60 s or until infants looked away for 2 s. Videos were used instead of static images to maintain infants' attention.

Each infant was habituated to an object with one of three possible skeletons in Experiment 1 and with one of two possible skeletons in Experiment 2 (see *Figure 1*), with half of the infants habituated to each surface form in each experiment. Infants met the habituation criterion when the average looking time in the preceding four trials was less than 50% of the average looking time in the first four trials. Test trials were presented after infants had habituated or following 24 habituation trials, whichever came first. All infants were presented with a total of six test trials, alternating between objects with the same or different skeletons (3 test trials of each type). The type of first test trial (same or different skeleton) was counterbalanced across infants.

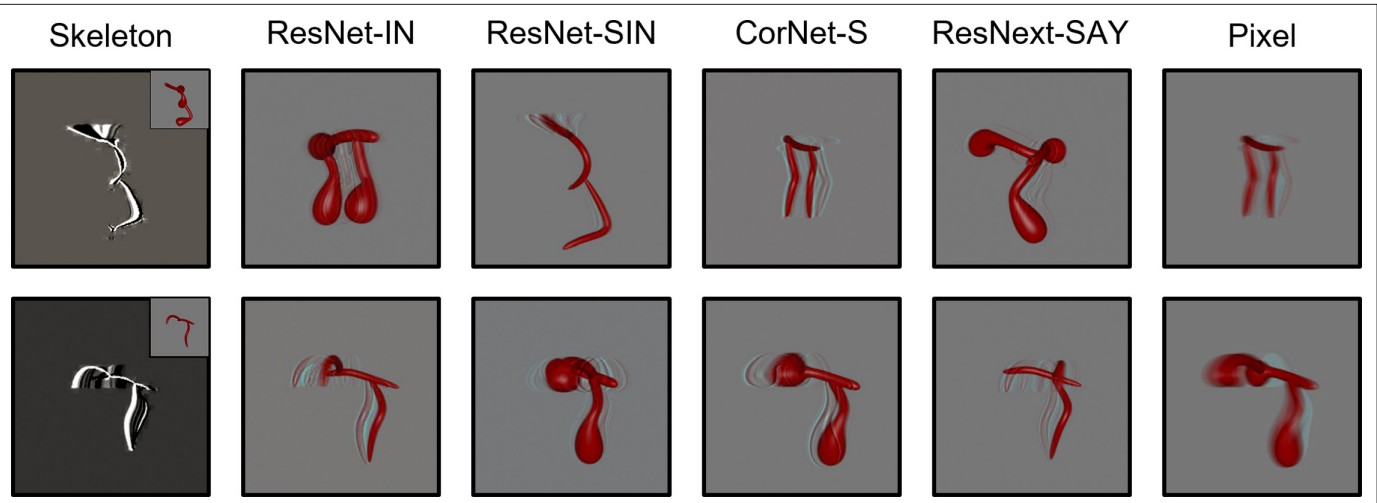

**Figure 4.** Examples of autoencoder reconstructions using objects from Experiment 1 (top) and Experiment 2 (bottom) for all models except FlowNet. FlowNet reconstructions are not possible because it requires multiple frames as input. For the Skeletal model, the inset displays the original input image. Each reconstruction was created by feeding a random frame from the habituation object video to each model immediately following its habituation to said video.

## Model descriptions

For our Skeletal model, we used a flux-based medial axis algorithm (*Dimitrov et al., 2003*; *Rezanejad and Siddiqi, 2013*) which computes a 'pruned' skeletal structure tolerant to local contour variations (*Feldman and Singh, 2006*). A pruned Skeletal model was selected for its biological plausibility in describing human shape judgments (*Ayzenberg et al., 2019a*; *Feldman et al., 2013*; *Wilder et al., 2011*; *Wilder et al., 2019*).

The four ANNs tested in the present study were: ResNet-IN, ResNet-SIN, CorNet-S, and ResNext-SAY. ResNet-IN and ResNet-SIN are 50-layer residual networks (*He et al., 2016*) chosen for their strong performance on object recognition tasks. ResNet-IN was trained on ImageNet (*Russakovsky et al., 2015*), a dataset consisting of high-quality naturalistic photographs, whereas ResNet-SIN was trained on Stylized-ImageNet, a variation on the conventional ImageNet dataset that decorrelates color and texture information from object images using style transfer techniques (*Geirhos et al., 2018*; *Huang and Belongie, 2017*). The third model, CorNet-S, is a 5-layer recurrent network explicitly designed to mimic the organization and functional profile of the primate ventral stream (*Kubilius et al., 2019*). It was chosen because it is a biologically plausible model of primate object recognition behavior and neural processing, as measured by the brain-score benchmark (*Schrimpf et al., 2018*). Finally, ResNext-SAY uses an updated version of the ResNet architecture and was designed to approximate the visual processing abilities of an infant (*Orhan et al., 2020*). It was trained using a self-supervised temporal classification method on the SAYCam dataset (*Sullivan et al., 2020*), a large, longitudinal dataset recorded from three infants' first-person perspectives.

To test whether infant-looking behaviors could be accounted for by low-level image properties, we also tested a model of pixel similarity and FlowNet, a model of motion flow (*Dosovitskiy et al., 2015*; *Ilg et al., 2017*). For the Pixel model, the raw image frame was passed into the evaluation pipeline and classification was based on this information alone. FlowNet estimates the motion energy between adjacent video frames and is able to successfully use motion information to segment objects from the background as well as support action recognition in videos. Here we used an iteration of FlowNet known as FlowNet2-S, which was pre-trained on the MPI-Sintel and 'flying chairs' datasets (*Butler et al., 2012*; *Mayer et al., 2016*).

## Model analyses

Models were evaluated on the same stimulus sets presented to infants and tested using methods similar to the habituation/dishabituation procedure. One way to conceive of this procedure is as a measure of alignment between the stimulus and the infant's internal representation of the stimulus.

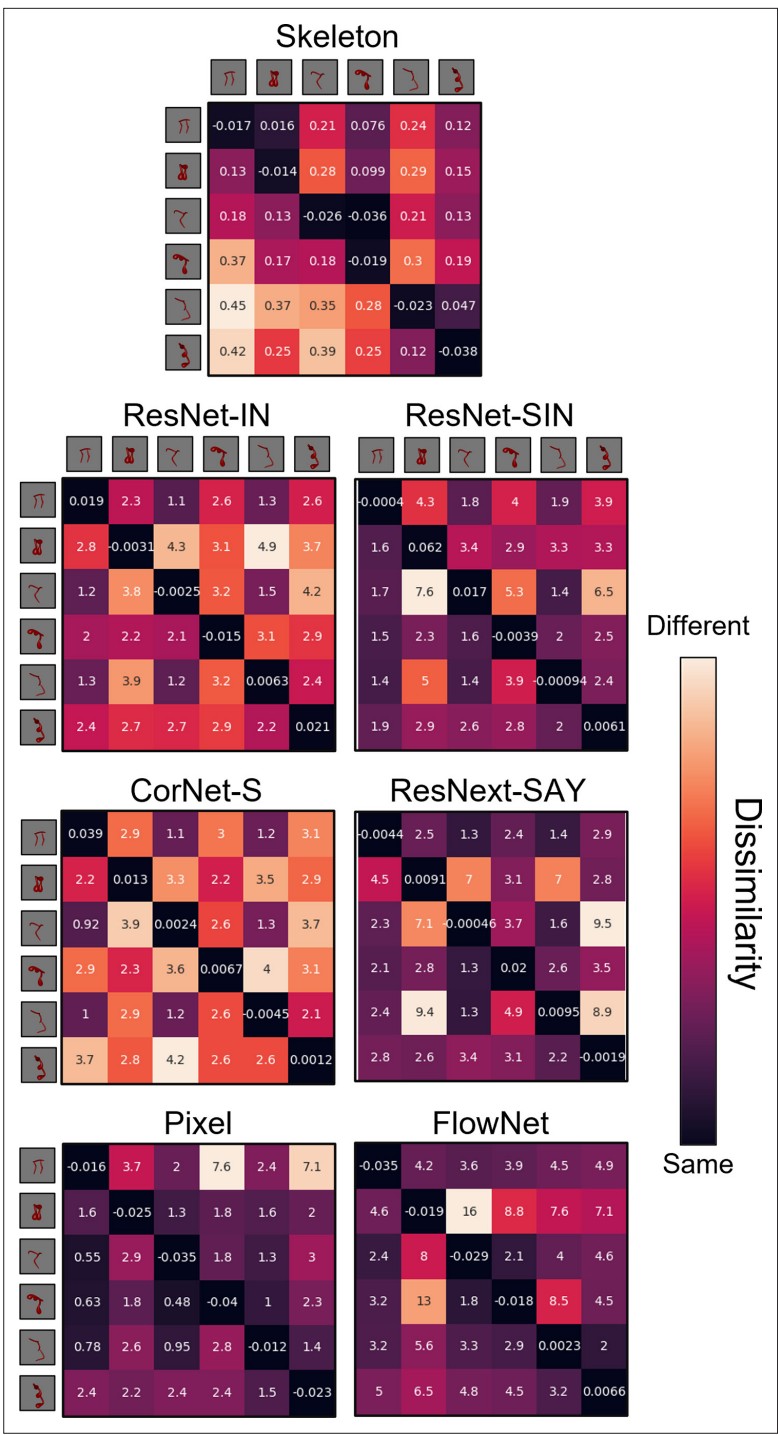

**Figure 5.** Dissimilarity matrices for each computational model in Experiment 1. Dissimilarity for each object pair was calculated as the error from an autoencoder following habituation to one object and testing on a second object. Internal values of each cell in the matrix indicate the error between habituation and test objects. Error-values are normalized to the end of habituation. Dissimilarity matrices are asymmetrical because the error value changes depending on which object the model was habituated to. The object adjacent to each row is the habituation object, and the object adjacent to each column is the test object.

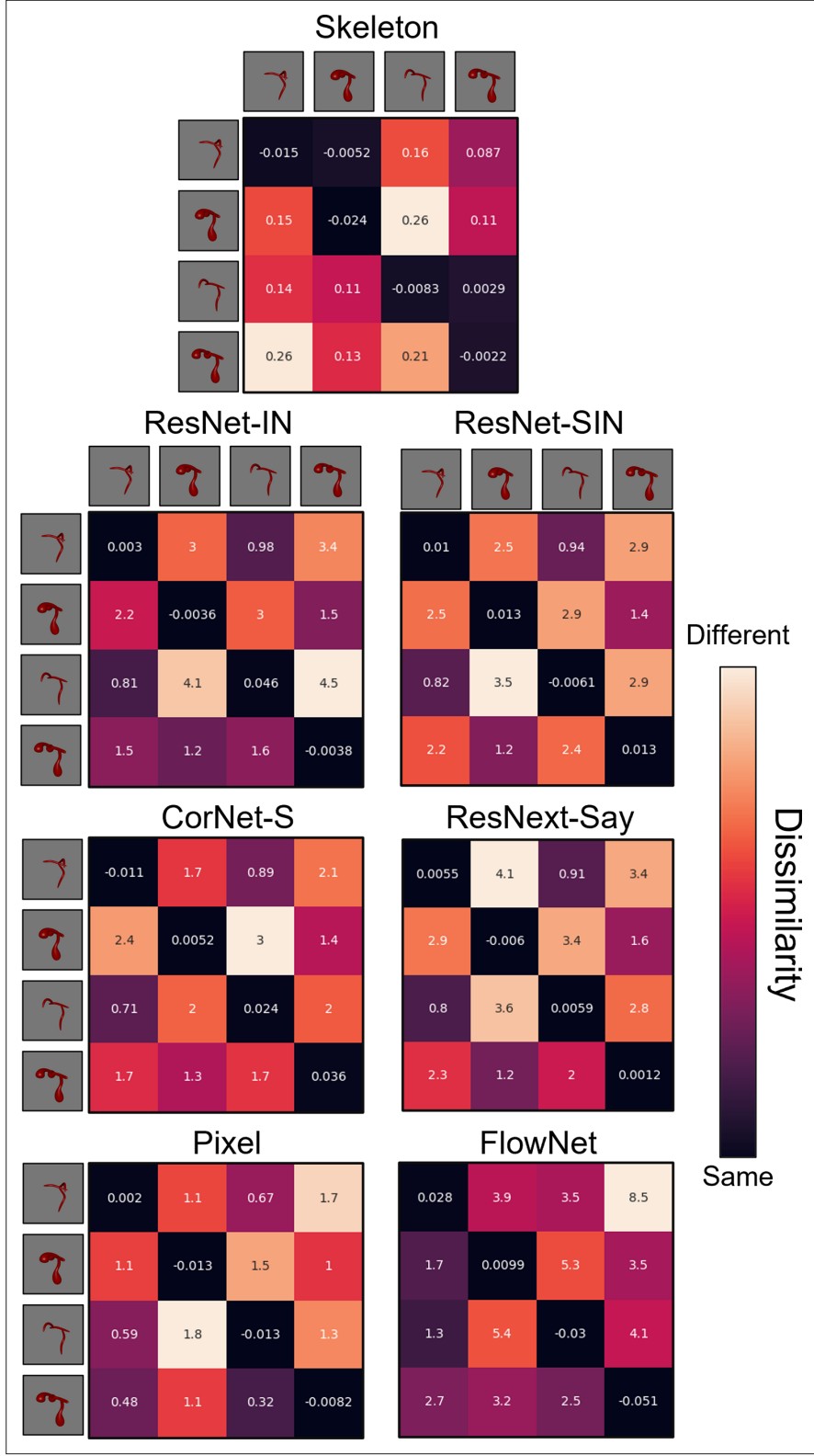

**Figure 6.** Dissimilarity matrices for each computational model in Experiment 2. Dissimilarity for each object pair was calculated as the error from an autoencoder following habituation to one object, and testing on a second object. Internal values of each cell in the matrix indicate the error between habituation and test objects. Error-values are normalized to the end of habituation. Dissimilarity matrices are asymmetrical because the error value changes depending on which object the model was habituated to. The object adjacent to each row is the habituation object, and the object adjacent to each column is the test object.

Infants will continue looking at a display for as long as they perceive a mismatch, or 'error', between the stimulus and their representation. To approximate this process, we converted each model into an autoencoder, which was 'habituated' and tested using the same criteria as infants (*Mareschal et al., 2000*; *Westermann and Mareschal, 2004*). An autoencoder is an unsupervised learning model that attempts to recreate the input stimulus using a lower-dimensional set of features than the input. Like infants, the error signal from the output layer of an autoencoder remains high when there is a mismatch between the input stimulus and the internal representation. Each model was converted into an autoencoder by passing the outputs of the model to a single transposed convolutional decoding layer. For ANNs, outputs were extracted from the penultimate layer of the model, 'AvgPool' (with ReLu) for each frame of the video. For the Skeletal model, each frame of the video was first binarized and the skeleton extracted. The resulting skeletal image was blurred using a 3-pixel Gaussian kernel and passed into the decoder. For FlowNet, image representations of the flow fields were generated for each pair of adjacent frames before being passed into the decoder. No image preprocessing was conducted for the Pixel model. To match the output dimensions of each ANN, each image from the Skeletal, FlowNet, or Pixel model was passed through a single convolutional feature extraction layer with Max and Average pooling, before being sent to the decoding layer.

During the habituation phase, models were shown repeated presentations (epochs) of an object. Habituation was accomplished by training the models on each video using the Adam optimizer and a mean squared error loss function. For the ANNs, the weights of the decoding layer were updated during habituation. The weights of the pretrained models were frozen. For the Skeletal, FlowNet, and Pixel models, the weights of both the output and decoding layers were updated to support efficient feature extraction. The Skeletal, FlowNet, or Pixel model backbone was not altered. Models were said to have habituated once the average error signal in the last four epochs was below 50% of the average error in the first four epochs. The Skeletal model and ANNs met the habituation criteria within 8 epochs. Pixel and FlowNet models met the habituation criteria within 22 and 43 trials, respectively. The Skeletal, ANN, and Pixel models showed good reconstruction of the habituated objects (see *Figure 4*). Reconstructions using FlowNet were not possible because multiple frames were used as input. At test, models were presented with objects that had the same/different skeletons or the same/different surface forms as the habituated object, and the error signal was recorded. See *Figures 5 and 6* for the error signal between all object pairs.

## Classification score

For comparison, the performance levels of infants and computational models were converted to a classification score. Organisms' responses (i.e. looking time/error signal) in the test phase were first normalized to the end of the habituation phase (last 4 trials/epochs) by taking the difference between the two. The response to the novel object was then converted into a proportion by dividing it by the combined response to the novel and familiar test object. For both the models and infants, a classification score of 0.50 reflects chance performance.

---

# Additional information

### Funding

| Funder | Grant reference number | Author |
| --- | --- | --- |
| National Institutes of Health | T32 HD071845 | Vladislav Ayzenberg |

The funders had no role in study design, data collection and interpretation, or the decision to submit the work for publication.

### Author contributions

Vladislav Ayzenberg, Conceptualization, Data curation, Formal analysis, Investigation, Methodology, Project administration, Validation, Visualization, Writing - original draft, Writing - review and editing; Stella Lourenco, Conceptualization, Methodology, Project administration, Resources, Supervision, Writing - original draft, Writing - review and editing

## Author ORCIDs
Vladislav Ayzenberg http://orcid.org/0000-0003-2739-3935
Stella Lourenco http://orcid.org/0000-0003-3070-7122

## Ethics

Human subjects: All families gave informed consent according to a protocol approved by the Emory University Institutional Review Board (IRB) under the project 'Spatial Origins' (Study Number IRB0003452).

## Decision letter and Author response

Decision letter https://doi.org/10.7554/eLife.74943.sa1
Author response https://doi.org/10.7554/eLife.74943.sa2

---

# Additional files

## Supplementary files

• Transparent reporting form

## Data availability

All stimuli and data are available at: https://osf.io/4vswu/.

The following dataset was generated:

| Author(s) | Year | Dataset title | Dataset URL | Database and Identifier |
|---|---|---|---|---|
| Ayzenberg V, Lourenco SF | 2022 | Perception of an object's global shape is best described by a model of skeletal structure in human infants | https://doi.org/10.17605/OSF.IO/4VSWU | Open Science Framework, 10.17605/OSF.IO/4VSWU |

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
