## [Editor Report]

This well-conducted study uses relatively large sample sizes, comprehensive statistical testing, and state-of-the-art modeling to provide novel evidence that human infants generalize shape from single examples on the basis of the "shape skeleton", a structural description of the part structure of the shape. It will be of interest to researchers working on object shape processing and on the development of visual perception.

---

## [Decision Letter]

**Decision letter after peer review:**

[Editors’ note: the authors submitted for reconsideration following the decision after peer review. What follows is the decision letter after the first round of review.]

Thank you for submitting the paper "The shape skeleton supports one-shot categorization in human infants: Behavioral and computational evidence" for consideration by *eLife*. Your article has been reviewed by 3 peer reviewers, one of whom is a member of our Board of Reviewing Editors, and the evaluation has been overseen by a Senior Editor. The reviewers have opted to remain anonymous.

We are sorry to say that, after consultation with the reviewers, we have decided that this work will not be considered further for publication by *eLife*.

Specifically, the reviewers thought that results may be alternatively explained by motion similarity and/or low-level visual similarity between habituation and test stimuli, as explained in more detail below.

*Reviewer #1:*

This is a very lucidly written article on a fascinating and important topic: how humans are able to learn novel visual categories based on few or even just one single example. This ability can be contrasted with conventional machine learning models which typically require massive data sets with thousands of training examples to learn the ability to categorize novel examples accurately. How humans generalize so well from such sparse training data remains poorly understood and working out how we achieve this is important not only for the psychological sciences, but also has implications for artificial intelligence and machine learning too.

The authors start with a strong premise, which is also likely to be true: a crucial aspect of human one-shot visual learning likely involves the perceptual processes that segment observed shapes into parts and represent them as a hierarchy of limbs, in a representation known in the field as 'shape skeletons'. There is a long history of evidence suggesting the human visual system analyses shape in this way, and such representations naturally lend themselves to abstractions and generalizations that are robust against significant variations in appearance, such as the surface structure and even pose of objects. While this is an excellent starting point for investigating one-shot learning, the idea that such representations might play a role in human visual categorization more generally is rather well accepted in the field, and thus perhaps not so original on its own. The potential novelty and importance of the contribution therefore rests on demonstrating that such representations are central to one-shot learning in particular.

Unfortunately, due to their choice of stimuli, I believe that the experiments the authors perform do not yet provide compelling evidence that the infants' looking times are driven by skeletal representations, or indeed whether one-shot learning is necessarily playing a role in their habituation.

The problem is that the different Surface Forms of each Skeleton are more similar to one another than those of the other Skeletons, in terms of the raw pixel similarity. I took screenshots of the stimuli from the MS and compared them in MATLAB. For 7 out of 10 possible comparisons, the corresponding Surface Forms from the same 'category' are closer in raw pixel terms than to any of the rivals. This leads me to believe that the pattern of results is based on the raw physical (and thus perceptual) similarity between the habituation and test stimuli, rather than based on one-shot generalization per se, or on skeletal representations of the shapes in particular. In my opinion, the fact that sophisticated artificial neural network models don't predict this is not in itself strong evidence against my suggested explanation of the findings.

In future studies, this issue could be addressed by using stimuli for which this confound is not a problem. This could be achieved, for example, by using additional transformations of the objects, such as rotations, that preserve part structure but radically alter the projected image of the objects. This way skeletal structure could be decoupled from straightforward image similarity, and a stronger case for generalization beyond the 'one-shot' training exemplar would be provided.

*Reviewer #2:*

This is an important paper. The issue of "one-shot" learning---how people can learn categories and concepts from a single example or set of examples---lies at the heart of the current controversies over "deep learning" models. These models are ubiquitous these days and are often promoted as descriptive models of human learning, but they inherently require many trials to learn. But human learners can often learn from single trials, presenting a fundamental challenge to the entire deep learning paradigm. This has been pointed out in broad terms before, but to really advance this debate, a study needs to establish that humans can indeed learn from single examples, simultaneously demonstrate that appropriate deep learning models cannot, and explain something about exactly what why---all of which this paper accomplishes with impressive rigor.

In that context, the paper specifically studies shape category learning by human infants, a particularly interesting case, and establishes a specific pattern: that infants generalize shape categories based on the "shape skeleton," a structural description of the part structure of the shape. That is, infants exposed to a single shape with a particular shape skeleton tend to be "unsurprised" by other shapes with the same skeleton, but more surprised by shapes with different skeletons, indicating that the single example was enough to establish in their minds an apparent shape category.

This generalization is in a sense unsurprising, because the shape skeleton is supposed to define characteristic "invariants" within shape categories, but it is nevertheless novel. Although the tendency for infants to generalize lexical categories based on shape is well established, the precise nature of one-shot shape generalizations has not previously been studied. For researchers in shape this is a very important result, and for anybody interested in how humans learn categories I think is is both fundamental and thought-provoking.

I found the methodology and statistical analysis in the paper comprehensive and rigorous, and the writing clear, so and have only relatively minor comments on the manuscript, which follow. I don't see page numbers, so give quotations to indicate the relevant location.

– "a phenomenon known as 'one-shot categorization'" – The issue of one-shot category learning, and the computational question of what makes it possible for human learners to instantly generalize from some examples but not others, has been studied somewhat more extensively than this very brief introduction lets on. I would point to for example Feldman (1997, J. Math Psych) which explicitly takes it up and argues for a mechanism related to the current paper (namely, that one-shot categorization is possible when the one example implies a highly specific structural model).

– "However, one might ask whether these findings are truly indicative of categorization, or simply better discrimination of objects with the different skeletons." I'm not sure these are really different explanations. Learners are better at discriminating objects that appear to be from different categories (categorical perception, etc.).

– "infants' looking times on the first test trial did not differ for within- and between-category test objects" – This claim is followed by a non-significant NHST test, which does not allow an affirmative conclusion of no difference here, and a Bayes Factor, which does. The NHST test really doesn't add anything. I personally think these tests could be omitted throughout the paper in favor of the more informative BFs – but particularly when null results are discussed.

– members from different -> members of different.

– "we tested models by feeding their outputs into an autoencoder and measuring the error signal across habituation and test phases (see Methods)." I didn't quite get this. To evaluate similarity within the network models, can't one use a Euclidean norm or a cosine? Please clarify.

– "but did not differ from one another" – Meaning what? Low BF? If so, please give BF.

– "it has remained unknown whether one-shot categorization is possible within the first year of human life." Has this really never been established, even for simple categories? Anecdotally, infants seem to do one-shot learning all the time.

– "Moreover, V2 and V3 are evolutionarily preserved in primate and non-primate animals" I think the entire discussion of neural analogs here is misleading. I am not a bird expert but I didn't think birds have homologous visual cortical areas to primates. But that doesn't matter, because functional organization is analogous when computational problems are analogous. In other words, birds don't generalize the same way we do because they have V2 areas like us, but because they are solving problems like us.

– "set was comprised of two…" -> "set comprised two…" Use of "comprised of" to mean "composed of" is colloquial. The US comprises states, not the other way around.

*Reviewer #3:*

This study relates habituation in infants to categorization in neural networks, showing that infants learn (as evidenced by looking times) shape skeletons across surface form, while neural networks that lack an explicit skeletal representation do not show this generalization. A key aspect of the study is that infants are only exposed to one exemplar, suggesting that infants learn shape skeletons using "one-shot categorization". The study is well-conducted, using relatively large sample sizes, comprehensive statistical testing, and careful modelling. The manuscript is well-written and easy to follow. However, results may be alternatively explained by motion similarity and/or low-level visual similarity between habituation and test stimuli.

– The shapes are shown as videos during both habituation and test phases. While I understand that this was preferred for drawing the infants' attention, it complicates the interpretation of the results. First, it becomes hard to disentangle the learning of the shape skeleton from the learning of the motion trajectory. Second, the comparison with neural networks becomes more difficult as these neural networks are not sensitive to motion.

– Because surface form differed for both test objects, the same-skeleton object will be visually more similar to the habituated object than the different-skeleton object. Therefore, it cannot be ruled out that results reflect habituation to lower-level stimulus properties rather than shape skeleton. This is also suggested by recent fMRI results using these stimuli (Ayzenberg et al., Neuropsychologia 2022), showing that the skeletal model correlates with activity patterns throughout the visual system, including V1 (though the cross-surface form results are only shown for V3 and LO, as far as I could tell).

I would recommend the authors to include a discussion of the possible contributions of motion and non-skeletal visual properties to the habituation results.

[Editors’ note: further revisions were suggested prior to acceptance, as described below.]

Thank you for choosing to send your work entitled "The shape skeleton supports one-shot categorization in human infants: Behavioral and computational evidence" for consideration at *eLife*. Your letter of appeal has been considered by a Senior Editor and a Reviewing Editor, and we are prepared to consider a revised submission with no guarantees of acceptance.

In addition to the comments of the previous reviews, during the consultation of the appeal, the following questions came up:

1. What new insight do we gain from the infant study above and beyond what we already know from adults?

2. Why do you think that (dis)habitation is evidence of learning per se, rather than comparisons of perceptual similarity between items?

[Editors’ note: further revisions were suggested prior to acceptance, as described below.]

Thank you for resubmitting your work entitled "The shape skeleton supports one-shot categorization in human infants" for further consideration by *eLife*. Your revised article has been evaluated by Floris de Lange (Senior Editor) and a Reviewing Editor, in consultation with reviewers.

The manuscript has been improved but there are some remaining issues that need to be addressed, as outlined below:

The reviewers were not convinced that the experiments convincingly demonstrate “one-shot” categorization. This concern can be addressed by adjusting the headline claim throughout (title, abstract, intro, results and discussion would need some modifications). A suggested alternative title could be: "The shape skeleton supports shape similarity judgments in human infants".

The design involves making a judgment about which objects appear to be more similar. This requires comparing the distances between presented items (in some feature space describing the stimuli). Making such a comparison doesn't involve learning anything on the basis of the experience. It doesn't involve generalization and there is no sense in which it is a 'one-shot' task, except that a given trial presents only a few items. But this is true of practically ANY experiment involving comparing a small number of items.

Here is an analogy: suppose the infants were shown three different (i.e., easily discriminable) patches of gray shades: two patches are relatively light shades of gray and one is a significantly darker shade. A habituation experiment like the one the authors performed would reveal that the infants see the two light grays as more similar than the darker one. But in what sense is this 'one shot categorization'? The infants wouldn't have learned a new category. There is no meaningful generalization. The experiment simply reflects the fact that similar grays look more similar than more different ones.

The same is true in the authors' experiments. The experiments demonstrate that objects with more similar skeletons appear more similar to infants. This is not a trivial result, and is worthy of publication in its own right (although similar findings have already been shown in adults). However not under the title of 'one-shot categorization'. Instead, it should be pitched (correctly) as the shape skeleton contributing substantially to judgments of shape similarity.

To reiterate: I think the authors have performed an elegant study with some interesting findings. I think their interpretation can certainly be discussed in the MS. However, the headline claim about one-shot categorization should be toned down. Otherwise, the term would 'one-shot' would not mean anything anymore.

---

## [Author Response]

[Editors’ note: The authors appealed the original decision. What follows is the authors’ response to the first round of review.]

Reviewer #1:This is a very lucidly written article on a fascinating and important topic: how humans are able to learn novel visual categories based on few or even just one single example. This ability can be contrasted with conventional machine learning models which typically require massive data sets with thousands of training examples to learn the ability to categorize novel examples accurately. How humans generalize so well from such sparse training data remains poorly understood and working out how we achieve this is important not only for the psychological sciences, but also has implications for artificial intelligence and machine learning too.The authors start with a strong premise, which is also likely to be true: a crucial aspect of human one-shot visual learning likely involves the perceptual processes that segment observed shapes into parts and represent them as a hierarchy of limbs, in a representation known in the field as 'shape skeletons'. There is a long history of evidence suggesting the human visual system analyses shape in this way, and such representations naturally lend themselves to abstractions and generalizations that are robust against significant variations in appearance, such as the surface structure and even pose of objects. While this is an excellent starting point for investigating one-shot learning, the idea that such representations might play a role in human visual categorization more generally is rather well accepted in the field, and thus perhaps not so original on its own. The potential novelty and importance of the contribution therefore rests on demonstrating that such representations are central to one-shot learning in particular.Unfortunately, due to their choice of stimuli, I believe that the experiments the authors perform do not yet provide compelling evidence that the infants' looking times are driven by skeletal representations, or indeed whether one-shot learning is necessarily playing a role in their habituation.The problem is that the different Surface Forms of each Skeleton are more similar to one another than those of the other Skeletons, in terms of the raw pixel similarity. I took screenshots of the stimuli from the MS and compared them in MATLAB. For 7 out of 10 possible comparisons, the corresponding Surface Forms from the same 'category' are closer in raw pixel terms than to any of the rivals. This leads me to believe that the pattern of results is based on the raw physical (and thus perceptual) similarity between the habituation and test stimuli, rather than based on one-shot generalization per se, or on skeletal representations of the shapes in particular. In my opinion, the fact that sophisticated artificial neural network models don't predict this is not in itself strong evidence against my suggested explanation of the findings.

We thank the reviewer for raising this concern. It is indeed the case that the test objects with the familiar skeleton exhibited greater image-level similarity to the habituation object than the test objects with the novel skeleton. Importantly, however, greater numerical similarity does not guarantee greater perceptual similarity, as raw pixel differences could be indiscriminable to humans. Thus, as in adult psychophysics, we ensured that both test objects were equally discriminable from the habituation object by infants. As discussed on pages 6 and 8, infants discriminated both test objects from the habituation object. To address this concern further, we also compared infants to a model based on pixel similarity (see pages 6-9, 18-20). Although the Pixel model performed above chance in both experiments, we found that infants (and the Skeletal model) outperformed the pixel model in both cases (see pages 7-9). Thus, despite some evidence that pixel similarity may allow for categorization, it is unlikely that this alone accounts for infants’ performance. Nevertheless, we now discuss the potential role of image-level similarity (and motion trajectory) to object categorization in the General Discussion (see page 11-12).

In future studies, this issue could be addressed by using stimuli for which this confound is not a problem. This could be achieved, for example, by using additional transformations of the objects, such as rotations, that preserve part structure but radically alter the projected image of the objects. This way skeletal structure could be decoupled from straightforward image similarity, and a stronger case for generalization beyond the 'one-shot' training exemplar would be provided.

We have provided greater justification for the claim that infants’ performance was based on the shape skeleton, rather than image similarity alone. Nevertheless, we agree with the reviewer that future research would benefit from the suggested manipulations.

Reviewer #2:This is an important paper. The issue of "one-shot" learning---how people can learn categories and concepts from a single example or set of examples---lies at the heart of the current controversies over "deep learning" models. These models are ubiquitous these days and are often promoted as descriptive models of human learning, but they inherently require many trials to learn. But human learners can often learn from single trials, presenting a fundamental challenge to the entire deep learning paradigm. This has been pointed out in broad terms before, but to really advance this debate, a study needs to establish that humans can indeed learn from single examples, simultaneously demonstrate that appropriate deep learning models cannot, and explain something about exactly what why---all of which this paper accomplishes with impressive rigor.In that context, the paper specifically studies shape category learning by human infants, a particularly interesting case, and establishes a specific pattern: that infants generalize shape categories based on the "shape skeleton," a structural description of the part structure of the shape. That is, infants exposed to a single shape with a particular shape skeleton tend to be "unsurprised" by other shapes with the same skeleton, but more surprised by shapes with different skeletons, indicating that the single example was enough to establish in their minds an apparent shape category.This generalization is in a sense unsurprising, because the shape skeleton is supposed to define characteristic "invariants" within shape categories, but it is nevertheless novel. Although the tendency for infants to generalize lexical categories based on shape is well established, the precise nature of one-shot shape generalizations has not previously been studied. For researchers in shape this is a very important result, and for anybody interested in how humans learn categories I think is is both fundamental and thought-provoking.I found the methodology and statistical analysis in the paper comprehensive and rigorous, and the writing clear, so and have only relatively minor comments on the manuscript, which follow. I don't see page numbers, so give quotations to indicate the relevant location.– "a phenomenon known as 'one-shot categorization'" – The issue of one-shot category learning, and the computational question of what makes it possible for human learners to instantly generalize from some examples but not others, has been studied somewhat more extensively than this very brief introduction lets on. I would point to for example Feldman (1997, J. Math Psych) which explicitly takes it up and argues for a mechanism related to the current paper (namely, that one-shot categorization is possible when the one example implies a highly specific structural model).

We thank the reviewer for this suggestion. In the general discussion, we now situate our results within the broader one-shot categorization literature and have included the suggested paper (see pages 12-13).

– "However, one might ask whether these findings are truly indicative of categorization, or simply better discrimination of objects with the different skeletons." I'm not sure these are really different explanations. Learners are better at discriminating objects that appear to be from different categories (categorical perception, etc.).

We agree with the reviewer that visual similarity plays an important role in categorization. However, an important prerequisite of categorization is that exemplars within a category are distinct from one another, yet grouped into the same category. Thus, and in response to concerns about image similarity, we provide evidence that the test objects were comparably discriminable from the habituation object (see pages 6 and 8).

– "infants' looking times on the first test trial did not differ for within- and between-category test objects" – This claim is followed by a non-significant NHST test, which does not allow an affirmative conclusion of no difference here, and a Bayes Factor, which does. The NHST test really doesn't add anything. I personally think these tests could be omitted throughout the paper in favor of the more informative BFs – but particularly when null results are discussed.

Though we appreciate the reviewer’s suggestion, we have opted to continue to include NHST to ensure accessibility to a larger community, who may be less familiar with Bayes factors. Importantly, though, all of the statistics converge on the same results.

– members from different -> members of different.

Thank you. We have made this correction (see pages 6 and 8).

– “we tested models by feeding their outputs into an autoencoder and measuring the error signal across habituation and test phases (see Methods).” I didn’t quite get this. To evaluate similarity within the network models, can’t one use a Euclidean norm or a cosine? Please clarify.

In the current study, we chose to use an autoencoder approach because it allowed us to better match evaluation of the models to infants (for review, see Yermolayeva & Rakison, 2014 – *Psychological Bulletin*). Specifically, unlike conventional measures of classification, autoencoders allow for measuring model categorization following exposure to just one exemplar, rather than abelled contrasting examples (e.g., supervised classifiers). Moreover, like infant learning during habituation, the learned representation of an autoencoder reflects the entire habituation video, rather than the similarity between individual frames, as might be measured by Euclidean or cosine measures. Most importantly, unlike other techniques, autoencoders can be tested using the same criteria as infants. We have provided further justification for the use of autoencoders on pages 6-7.

– “but did not differ from one another” – Meaning what? Low BF? If so, please give BF.

The lack of a difference is based on overlapping confidence intervals (see page 7).

– “it has remained unknown whether one-shot categorization is possible within the first year of human life.” Has this really never been established, even for simple categories? Anecdotally, infants seem to do one-shot learning all the time.

To the best of our knowledge, this is the first study to test one-shot categorization in human infants. A predominant perspective in developmental psychology is that one-shot categorization on the basis of shape is possible at around 4 years of age following more extensive linguistic and object experience (see page 3 and 12).

– “Moreover, V2 and V3 are evolutionarily preserved in primate and non-primate animals” I think the entire discussion of neural analogs here is misleading. I am not a bird expert, but I didn't think birds have homologous visual cortical areas to primates. But that doesn't matter, because functional organization is analogous when computational problems are analogous. In other words, birds don't generalize the same way we do because they have V2 areas like us, but because they are solving problems like us.

We have removed the aforementioned section from the General Discussion.

– "set was comprised of two…" -> "set comprised two…" Use of "comprised of" to mean "composed of" is colloquial. The US comprises states, not the other way around.

We have made this correction (see page 13).

Reviewer #3:This study relates habituation in infants to categorization in neural networks, showing that infants learn (as evidenced by looking times) shape skeletons across surface form, while neural networks that lack an explicit skeletal representation do not show this generalization. A key aspect of the study is that infants are only exposed to one exemplar, suggesting that infants learn shape skeletons using "one-shot categorization". The study is well-conducted, using relatively large sample sizes, comprehensive statistical testing, and careful modelling. The manuscript is well-written and easy to follow. However, results may be alternatively explained by motion similarity and/or low-level visual similarity between habituation and test stimuli.– The shapes are shown as videos during both habituation and test phases. While I understand that this was preferred for drawing the infants' attention, it complicates the interpretation of the results. First, it becomes hard to disentangle the learning of the shape skeleton from the learning of the motion trajectory. Second, the comparison with neural networks becomes more difficult as these neural networks are not sensitive to motion.

The reviewer raises a valid concern. However, we would point out that the skeletal model, like the neural networks, is not sensitive to motion, yet it performed comparably to infants, suggesting that motion information is not needed for models to succeed on one-shot categorization. We would also note that the neural networks were successful at categorizing objects by their surface forms from the same videos (see pages 9-10), suggesting that their deficit was related to the type of information used for categorization, not to the use of videos *per se*. Finally, we now include a model of motion flow (FlowNet), which we compare to infants. Although FlowNet performed above chance in both experiments, our data nevertheless suggest that the skeletal model was a better match to infants’ performance (see pages 6-9). Nevertheless, we now explicitly discuss the potential contribution of motion trajectory in forming object representations (see pages 11-12).

– Because surface form differed for both test objects, the same-skeleton object will be visually more similar to the habituated object than the different-skeleton object. Therefore, it cannot be ruled out that results reflect habituation to lower-level stimulus properties rather than shape skeleton. This is also suggested by recent fMRI results using these stimuli (Ayzenberg et al., Neuropsychologia 2022), showing that the skeletal model correlates with activity patterns throughout the visual system, including V1 (though the cross-surface form results are only shown for V3 and LO, as far as I could tell).

Please see our previous response (#3). Although it is true that the same-skeleton object was more similar than the different-skeleton object, in raw pixels, to the habituated object, infants’ looking times suggest that both test objects (same-skeleton and different-skeleton) were perceptually discriminable from the habituated object (see pages 6 and 8). Moreover, we also demonstrate that infants (and the Skeletal model) outperformed a model based on image-level similarity (Pixel model; see pages 6-9). Nevertheless, we now elaborate on the potential contribution of image-level similarity to categorization (see page 11).

Finally, we would note that although Ayzenberg et al. (2022) found that the Skeletal model was correlated with the response profile of V1, it did not explain unique variance in this region after controlling for low-level visual similarity. By contrast, the Skeletal model explained unique variance in areas V3 and LO, even when controlling for several other models of visual similarity, and even when surface forms varied. These findings suggest that there can be some covariation between the Skeletal model and other visual properties but, importantly, different models explain unique variance in human behavioral judgments and neural responses. These findings further highlight the importance of the current approach in which we compared infants to several models of vision, not only the Skeletal model.

I would recommend the authors to include a discussion of the possible contributions of motion and non-skeletal visual properties to the habituation results.

We thank the reviewer for their recommendation. We now include discussion of the possible contributions of motion and non-skeletal visual properties (see pages 11-12).

[Editors’ note: what follows is the authors’ response to the second round of review.]

In addition to the comments of the previous reviews, during the consultation of the appeal, the following questions came up:1. What new insight do we gain from the infant study above and beyond what we already know from adults?

Although adults are capable of one-shot categorization, this says nothing of its developmental origins. In particular, because of adults’ extensive experiences, it is difficult to disambiguate the contributions of early-developing perceptual mechanisms, supervised object experience, and language. With infants, we side-step these issues. By testing children who are largely non-verbal and who have had minimal experience with different objects, we provide a unique test of the perceptual mechanisms that support one-shot categorization. Moreover, few studies have compared the categorization abilities of infants to different computational models. This approach allows us to better assess the mechanism(s) underlying one-shot categorization and provides a novel avenue by which to evaluate the biological plausibility of existing models. Indeed, a common perspective in the machine learning and neuroscience literatures is that ANNs may simply need more extensive training to match human performance. Our study provides empirical evidence against this perspective by showing that one-shot categorization can be achieved early in human development, when language and object experiences are limited. Thus, by exploring object categorization at this developmental timepoint, we elucidate the initial processes that may support object categorization throughout the lifespan and provide a strong constraint on the development of object recognition models. In the revised manuscript, we have further elaborated on our motivation to test infants (see page 3).

2. Why do you think that (dis)habitation is evidence of learning per se, rather than comparisons of perceptual similarity between items?

We would argue that category learning and comparisons of visual similarity are not mutually exclusive. In particular, there is strong evidence that both children and adults make category decisions on the basis of visual similarity and that basic-level categories are grouped on the basis of visual similarity (e.g., Sloutsky, 2003 – Trends in Cognitive Sciences). The key question of our work is: what kind of visual properties are infants using to determine similarity, and, consequently, group membership? Habituation paradigms provide an excellent method to address this question because they reveal what properties, from the habituation phase, infants generalize to the test phase. Because there was no point at which habituation and test stimuli were presented simultaneously, infants’ looking behaviors were based on what they retained (i.e., learned) from the habituation phase, not direct comparisons of visual information.

We would also note that extensive research with infants has shown that their looking times in the test phase reflect the learned regularities of the habituation phase, such that, like adults, their category boundaries change depending on the variability of the stimuli during habituation (e.g., Bomba & Siqueland, 1983 – Journal of Experimental Child Psychology). This work suggests that infants learn the relevant properties of a category from the habituation phase. We have now provided additional review of the habituation and infant categorization literatures (see page 4).

[Editors’ note: what follows is the authors’ response to the second round of review.]

The manuscript has been improved but there are some remaining issues that need to be addressed, as outlined below:The reviewers were not convinced that the experiments convincingly demonstrate “one-shot” categorization. This concern can be addressed by adjusting the headline claim throughout (title, abstract, intro, results and discussion would need some modifications). A suggested alternative title could be: "The shape skeleton supports shape similarity judgments in human infants".The design involves making a judgment about which objects appear to be more similar. This requires comparing the distances between presented items (in some feature space describing the stimuli). Making such a comparison doesn't involve learning anything on the basis of the experience. It doesn't involve generalization and there is no sense in which it is a 'one-shot' task, except that a given trial presents only a few items. But this is true of practically ANY experiment involving comparing a small number of items.Here is an analogy: suppose the infants were shown three different (i.e., easily discriminable) patches of gray shades: two patches are relatively light shades of gray and one is a significantly darker shade. A habituation experiment like the one the authors performed would reveal that the infants see the two light grays as more similar than the darker one. But in what sense is this 'one shot categorization'? The infants wouldn't have learned a new category. There is no meaningful generalization. The experiment simply reflects the fact that similar grays look more similar than more different ones.The same is true in the authors' experiments. The experiments demonstrate that objects with more similar skeletons appear more similar to infants. This is not a trivial result, and is worthy of publication in its own right (although similar findings have already been shown in adults). However not under the title of 'one-shot categorization'. Instead, it should be pitched (correctly) as the shape skeleton contributing substantially to judgments of shape similarity.To reiterate: I think the authors have performed an elegant study with some interesting findings. I think their interpretation can certainly be discussed in the MS. However, the headline claim about one-shot categorization should be toned down. Otherwise, the term would 'one-shot' would not mean anything anymore.

Thank you for the additional opportunity to revise our manuscript. As suggested, we have changed the title of our manuscript to reflect the focus on shape similarity as opposed to one-shot categorization. The new title is: " Perception of an object’s global shape is best described by a model of skeletal structure in human infants." And we have revised the manuscript accordingly to reflect the new focus. The potential implications for one-shot categorization are now exclusively discussed in the general discussion.